# ADHD Symptomatology, Executive Function and Cognitive Performance Differences between Family Foster Care and Control Group in ADHD-Diagnosed Children

**DOI:** 10.3390/children8050405

**Published:** 2021-05-17

**Authors:** María Peñarrubia, Ignasi Navarro-Soria, Jesús Palacios, Javier Fenollar-Cortés

**Affiliations:** 1Department of Psychology, Universidad Loyola Andalucía, Avda. de las Universidades s/n, Dos Hermanas, 41704 Seville, Spain; mgpenarrubia@uloyola.es (M.P.); jfenollar@uloyola.es (J.F.-C.); 2Department of Developmental and Educational Psychology, University of Alacant, Ctra. Sant Vicent del Raspeig s/n, San Vicente, 03690 Alacant, Spain; 3Department of Developmental and Educational Psychology, University of Seville, C/Camilo José Cela s/n, 41018 Seville, Spain; jp@us.es

**Keywords:** attention deficit and hyperactivity disorder (ADHD), executive function, intellectual capacity, foster care, placement variables

## Abstract

Children in foster care have a high prevalence of attention deficit and hyperactivity disorder (ADHD) diagnosis, together with other difficulties in inattentive/hyperactive behaviors, executive and cognitive processes. Early exposure to adversity is a risk factor for developing ADHD via neurodevelopmental pathways. The goal of this research is (a) to study the cognitive and executive performance and inattentive/hyperactive behavior of ADHD-diagnosed children living in foster families in Spain, and (b) to analyze the role of placement variables in their performance. The sample was composed of 102 ADHD-diagnosed children aged 6- to 12-years-old, divided into two groups: 59 children living with non-relative foster families and 43 children not involved with protection services. Children’s executive function–inhibition, working memory, flexibility, attention, intellectual capacity, verbal comprehension, perceptive reasoning, working memory and processing speed were assessed using objective testing measures. At the same time, parents and teachers reported on children’s inattentive, hyperactive and impulsive behaviors. Children in foster care obtained lower scores in the general ability index than the control group after controlling the age at assessment. However, no differences were found in executive processes. Regarding placement factors, children with shorter exposure to adversities in their birth families and more time in foster care showed better executive performance. Professionals should consider the placement history of children in foster care and its influence on their symptomatology and cognitive capacities.

## 1. Introduction

Many children in the world are unable to live with their biological family due to various factors, including inadequate parenting style, neglect, and abuse [1]. In Spain, children declared by child protective services to be in a situation of significant abuse and neglect in their birth families are placed in out-of-home care. The first alternative is a foster family, although children often spend some time in residential care before being placed with a foster family. Foster families that may or may not be related to the child are declared suitable after going through interviews and training sessions. According to the last available statistics, in 2019, the Spanish state had the custody of almost 42,000 minors, of which 55% were in residential care (with a high proportion of children above 10 years), and the rest in family care [2]. However, family alternatives, such as fostering and adoption, imply different beneficial effects for children in need of out-of-home care [3] compared to the detrimental impact on developing institutional rearing [4]. Although foster and adoptive parents strive to offer children a protective, warming and loving context to overcome developmental delays and difficulties [5], they are confronted with challenges and difficulties [6].

Children in foster care are exposed to experiences of early adversity, including deprivation of certain experiences needed for normal development and exposure to threatening practices [7]. While in their birth families, they were exposed to neglect and different types and levels of abuse. Often, when foster care is not possible, or before foster care placement, they are placed in residential care, which inevitably deprives children of reciprocal and responsive interaction with a stable caregiver [8]—who, among other resources, provides a scaffolding of attention and stress regulation. The acute and chronic activation of stress mechanisms may produce long-term deficits in the structure and performance of the brain, especially when they occur in sensitive periods of development [9,10], including brain alterations on neuroanatomical, functional, metabolic and neurochemical activity [11,12,13]. These alterations can explain the deleterious and long-lasting consequences across several domains of functioning, including symptomatology related to attention deficit and hyperactivity disorder (ADHD from now on), executive and cognitive processes.

Hence, children in foster care are more likely to show symptoms related to ADHD such as impulsivity, overactivity and inattention difficulties [14,15,16,17]. The overall prevalence of ADHD diagnosis in children in foster care has been reported to range from 10% to 21% [18]. In large epidemiologic studies in the USA, the overall prevalence of ADHD in children in foster care was around 17%, a considerably higher rate than the general population, estimated at around 3.4% [19]. Similarly, in Spanish residential centers–where children usually spend some time before their arrival to foster care—the prevalence of ADHD was determined to be 17.9% [20]. Compared with other mental disorders, ADHD was the most frequent diagnosis in foster care children aged 3–11 years old [21]. In addition, children in foster care diagnosed with ADHD are more likely to receive multiple psychopharmacological agents than other ADHD-diagnosed populations [22].

Both inherited, and non-inherited factors contribute to developing ADHD. In addition to some gene variants, low birth weight/prematurity and pre and postnatal exposure to negative influences, including sometimes early exposure to extreme adversity, are considered risk factors for developing ADHD. However, it has not yet been found that any of them can cause ADHD on their own [23]. On one hand, children in foster care have been exposed to certain forms of early adversity that could be a risk factor for developing behaviors associated with ADHD due to the neurobiological alterations involved. On the other hand, exposure to early stressors, such as abuse or neglect, can exacerbate difficulties in processes, such as behavioral and attention regulation, which could overlap or mimic ADHD symptoms [24]. The combined action of both influences could explain the higher prevalence of ADHD diagnosis among children in foster care. However, the number of correct diagnoses can be erroneously magnified by the incorrect identification of mimetic symptoms.

In addition to a higher prevalence of ADHD, children in foster care frequently show difficulties in other aspects of executive function (EF from now on). Research findings, however, are not always homogeneous. Preschool children in foster care performed worse in cognitive flexibility assessed with neuropsychological tasks [25]. Opposite results were found by Pears and Fisher [26]. They did not find EF difficulties in the neuropsychological evaluation of preschool children in foster care. Scarce research has analyzed EF in Spanish children in foster care. A previous study with 43 Spanish foster children aged between five and nine used the behavior rating inventory of executive functions (BRIEF) [27], a caregiver-report questionnaire on everyday EF difficulties. Foster children obtained difficulties in most EF areas, especially in behavioral regulation, and 22% of the children had difficulties of clinical significance [28].

Previous research also has found difficulties in general cognitive function in foster care children [26,29,30]. Compared with community comparison groups, they have shown lower full-scale intelligence quotients, verbal comprehension and processing speed but average scores in working memory and perceptual reasoning [31]. The study conducted by Berger [32], who used multiple adjustments for selection factors that could influence child outcomes, ascertained that out-of-home placement did not worsen children’s cognitive abilities. Still, neither did it contribute to their improvement. Finally, specific studies with Spanish children in foster families showed intelligence quotient scores within a normal range [28].

Another influential factor on the ill development and subsequent recovery of children in foster care is the timing of the adversity. The age when adverse experiences occur can affect brain development, especially in regions with a more protracted trajectory that is more vulnerable to stress exposure [33]. Therefore, experiences of adversity during early childhood could imply bigger neurodevelopmental risks than adversity at a later age. It is more likely that children have not had the necessary experiences to ensure correct development [34]. However, several studies on children in foster care have not identified significant relations between EF and age at first placement [26], nor length of stay in residential care [20]. Only an older age at admission into the residential center was related to better performance in inhibition tasks [35], and an earlier age at placement in the adoptive family was related to fewer reported EF difficulties [36,37]. These contradictory results indicate the need to increase knowledge about children in foster care to explain the relationship between placement variables and later psychological development.

In conclusion, higher rates of ADHD symptoms and diagnosis prevalence are common in children in foster care, a neurodevelopmental disorder related to well-known patterns of executive and intellectual impairment. Children in foster care are more likely to have experienced adversities that contribute to developing behavioral and attention regulation difficulties that may exacerbate or mimic ADHD-like symptoms -including post-traumatic stress disorder, language and learning problems, reactive attachment disorder, and mood and anxiety disorders [24,38]. The current state of research leads us to evaluate whether the intellectual and executive profile and ADHD symptomatology in foster children diagnosed with ADHD is similar to the profile of a control group diagnosed with ADHD. In addition, we consider the relationship between placement variables and the children’s developmental profile and whether these variables contribute to exacerbate or buffer the results obtained by children in foster care. Although previous literature has explored the epidemiology and etiology of ADHD in foster children, this study aims to delve into the cognitive abilities developed by foster children diagnosed with ADHD. In this way, it may enhance a better understanding of these children’s needs in terms of assessment and intervention.

The specific goals of this study were (1) to analyze the ADHD symptomatology, EF performance and cognitive profile differences between foster care and a control group of children ADHD-diagnosed in Spain; and (2) to study the relation between the placement history and ADHD symptomatology, EF performance and cognitive profile in the group of children in foster care. Due to the exposure to early experiences of adversity, the accumulation of risk factors and the consequent alterations of the neural substrate, we expect that (1) compared with a control ADHD group, the ADHD-diagnosed foster group will show higher ADHD symptomatology—inattention, hyperactivity and impulsivity, a slight affectation of intellectual performance, and more global impairment in EF—especially in attention and inhibition. We predict that (2) a shorter stay in residential care, earlier age at placement and a longer time in the foster care family will be related to lower ADHD symptomatology, as well as better EF and intellectual performance.

## 2. Materials and Methods

### 2.1. Participants

Participants were 102 Spanish children (79.4% male) diagnosed with ADHD, aged between 6 and 12 years old (*M* = 8.29, *SD* = 1.92). Of these, 59 children (foster group; 76.7% male; age, *M* = 8.29, *SD* = 1.92) were in non-relative foster families, and 43 children (control group; 79.1% male; age, *M* = 8.30, *SD* = 1.96) were growing up with their biological families with no relation to child protection services. For the entire sample, 41 children (40.2%) met the criteria for ADHD, predominantly inattentive (ADHD-I), and 61 children (59.8%) met the criteria for ADHD-combined (ADHD-C). Given the scarce number of participants of the original clinical sample diagnosed with predominantly hyperactive/impulsive ADHD (ADHD-H), together with doubts about the temporal validity of the ADHD nominal subtype [39], cases with this diagnosis were not included in this study.

Children in foster care had lived with their biological families until they were an average age of 5.2 years (*SD* = 1.48) when they entered into residential care, where they stayed an average of 15.5 months (*SD* = 13.8). Children were placed into foster families at an average age of 6.49 years (*SD* = 2.00) and, at the time of the study, had been placed for an average of 22.2 months (*SD* = 10.7). The entire sample of the foster group experienced some type of abuse and/or neglect–according to the information obtained in the children’s files– when they were living with their biological families, although these variables were not included in further analysis.

The foster families were contacted through the program for non-relative foster families of Alacant (Spain). The control group was recruited using a simple random sampling in six public schools of the province of Alacant, assigned to the Children and Youth Mental Health Unit of the General University Hospital of Alacant. One inclusion criterion for the control group was that they lived with their biological family without previous contact with social services. There were several inclusion criteria for both groups: (i) to be aged 6 to 12 years old; (ii) have a prior diagnosis of ADHD by the public mental health system; (iii) have more than six symptoms in the ADHD rating scale-IV [40] in both forms, family and teachers (following ADHD DSM criteria); (iv) obtain a score higher than 80 in the general ability index (WISC-IV); (v) not obtain high-risk scores in disorders other than ADHD measured by the child and adolescent behavior inventory (CABI) [41]; and (vi) to be medication free for at least 48 hours before the time of the evaluation. Because the general ability index is more sensitive than the intelligence quotient to executive dysfunctions caused by ADHD [14,42], scores under 80 were excluded. The original sample consisted of 134 participants, but 32 children were excluded from the study: 23 children were under medication at the moment of assessment; eight participants did not meet ADHD criteria. Finally, one child did not meet the placement history criteria. Subjects with missing data were not included.

### 2.2. Measures

#### 2.2.1. EF Performance

The battery of neuropsychological assessment for executive function in children (ENFEN) is a Spanish battery of tests designed to assess EF performance from 6 to 12 years [43]. It includes four tasks, but the Rings task, a modification of the Hanoi Towers test to assess planning, was not used in this study due to the families’ time constraints and the research prioritization of other psychological domains. The other three tasks were always applied in the same order, with an administration time of approximately 20 min. These tasks allow assessing several executive processes: working memory as the ability to temporarily maintain, select, manipulate and transform information mentally without it being perceptively present [44]; inhibition capacity to contain a predominant motor response in a specific situation [45]; executive attention ability to suppress attention to irrelevant stimuli and selectively attend to chosen stimuli [46]; and flexibility readiness to change the concept system selectively in response to environmental stimuli [47]. Lastly, a total score for each task was obtained, and raw scores were transformed into sten scores (*M* = 5.5, *SD* = 2). ENFEN has been validated and scored for Spain’s child population. However, the official manual does not report reliability and validity data. Each task is independent of the other—they load in different factors, so removing one task has little effect on the others. In Maldonado [48], Cronbach’s alpha coefficient for the entire battery was 0.572; however, when the Inhibition task score was removed, it increased to 0.714.

Fluency task is an indirect measurement of working memory. The child has one minute to produce as many words as possible from both phonemic and semantic categories;Trail Making test assesses flexibility, inhibition, working memory and executive attention. In the gray trial, the child had to draw a line linking numbers from 20 to 1, which appeared randomly on a sheet of paper assessing working memory and executive attention. In the color version, the child must link numbers from 1 to 21, but he/she must switch between yellow and pink colors assessing to larger extent flexibility and inhibition;An inhibition task derived from the Stroop test assesses cognitive inhibition. A paper sheet shows three columns with 13 words in each. The words are color names printed with random color inks, but the color name and the color ink never match. The child must say the color ink of each word.

#### 2.2.2. Intellectual Capacity

The Wechsler Intelligence Scale for Children Fourth Edition (WISC-IV) [49] was used to assess general intellectual functioning. WISC-IV was standardized in Spain, showing adequate psychometric properties (WISC-IV Spanish) [50]; the modifications made from the English version were marginal. The standardization sample was representative of the Spanish population and included 1590 children aged 6–16 years stratified by age, gender, and region of the country. Furthermore, the structural validity of the Spanish version has shown adequate psychometric properties in children with ADHD [51]. As with versions for other countries, its core battery includes 10 subtests (*M* = 10, *SD* = 3) that contribute to the computation of a full-scale intelligence quotient, as well as four-factor index scores (*M* = 100, *SD* = 15): verbal comprehension, perceptive reasoning, working memory, and processing speed. Lastly, the general ability index (GAI) and cognitive processing index (CPI) were calculated. The GAI is based on verbal comprehension and perceptive reasoning indices, and the CPI summarizes performance on the working memory and processing speed indices. Administration time was approximately two hours, following the same order for applying the 10 subtests, and raw scores were converted into age-corrected standardized scores.

#### 2.2.3. ADHD Symptomatology and Diagnosis

The ADHD rating scale-IV [40] is an 18-item scale based on DSM-IV diagnostic criteria for ADHD. Family and teachers rated the occurrence of inattentive and hyperactivity/impulsivity symptoms for each child (nine items in each scale) for the past six months on a four-point scale (0 = never or rarely; 1 = sometimes; 2 = often; 3 = very often). Each item assesses a different symptom, and a score of 2 or 3 is considered an indication of the presence of the symptom. In each scale, the sum of the symptoms presented was used as the dependent variable, ranging between zero and nine. This scale has demonstrated adequate psychometric properties in studies with Spanish samples [52]. In the Spanish validation, Cronbach’s alphas for inattention and hyperactivity/impulsivity symptoms for family were α = 0.90 and α = 0.86, respectively, with the values being α = 0.95 and α = 0.94 for teachers, respectively. For the current study, following the DSM model, we assumed a categorical approach (that is, the number of symptoms instead of the sum of the scores).

Furthermore, the medical team of the Child and Youth Mental Health Unit carried out semi-structured interviews with the families to test whether the children met the DSM-IV criteria for ADHD [53]. When the clinical team thought it was appropriate, an interview with tutors and specialists involved in the intervention with the subject at school was also carried out.

### 2.3. Procedure

The Foster Care Intervention Program in the Province of Alacant (Spain) implements and supervises psychological intervention with foster families. This service contacted the families that met the inclusion criteria stipulated in the study, achieving the participation of all contacted families. Each family signed their consent and commitment to collaborate in training and research processes to improve the foster care resource. Families did not receive any compensation for participating in this study. The evaluation process of each child was carried out by the same psychologist during three sessions: file study, psychometric evaluation and family interview. Once the Directorate General for Equality and Inclusive Policies and the participating families gave their authorization, the information from the file of the minors was extracted. The retrieved information included: date of birth, date and reason for the declaration of protection order, start date and duration of residential care, start date and duration of family foster care. The foster families and the children were called to the intervention team’s premises, where the psychological evaluation of the children was carried out in a session of approximately two hours. The family interview took place in the family home in the course of one of the follow-up visits.

This research was conducted following the ethical principles included in the Declaration of Helsinki in its current form. The approval of the study and its methodology was granted by the institutional review board, the ethics committee of the University of Alacant and Directorate-General of Equality and Inclusive Policies, the agency responsible for the minors in the foster care program. Both foster parents and biological parents from the control group signed a written consent form, approved by the Ethics Committee of the University of Alacant (file number UA-2018-03-08). 

### 2.4. Analytic Strategy

Three different analyses were conducted. The first analysis aimed to contrast the average scores in ADHD symptomatology and intellectual and executive performance for foster care and control groups. Effect size analyses were also conducted, specifically Cohen’s *d* for parametric analyses and rank-biserial correlation for nonparametric analyses. A value of Cohen’s *d* of *d* = 0.2, *d* = 0.5 and *d* = 0.8 correspond to small, medium and large effects, respectively [54]. The second analysis explored the relations between ADHD symptoms, EF performance, intellectual capacity and placement history using correlation analyses. A correlation of *r* = 0.10 is considered a small effect, *r* = 0.30 a medium effect, and *r* = 0.50 a large effect [55]. Furthermore, multiple linear regression analyses were conducted to predict the intellectual and executive performance of the children in foster care based on ADHD symptoms and placement variables.

## 3. Results

Before conducting the analysis, data distribution was explored. The variables related to ADHD symptoms, ENFEN indexes and age, were all non-normally distributed (Shapiro–Wilk tests, *ps* < 0.05). Therefore, nonparametric tests were conducted in those cases.

### 3.1. Differences between Foster Care and Control Groups for ADHD Symptoms, EF Performance and Cognitive Profile

There were no differences between groups neither by age (*U* = 1265; *p* = 0.981) nor gender (χ^2^ = 0.005; *p* = 0.942). Likewise, no differences between groups were found in most of the variables (Table 1), except for “inattention” in the teacher form (foster group, *M* = 7.64, *SD* = 1.14; control group, *M* = 7.00, *SD* = 1.23; *U* = 1600; *p* = 0.018; *r* = 0.26), and “general ability index” (Foster group, *M* = 99.7, *SD* = 12.2; control group, *M* = 105.0, *SD* = 12.5; *t* = −2.126; *p* = 0.036; *d* = −0.43). In both cases, effect sizes were small to medium. As expected, “hyperactivity/impulsivity symptoms” variable was negatively correlated with “age” (ϱ = −0.43, *p* < 0.001, and ϱ = −0.31, *p* = 0.001, teacher and family form, respectively). Furthermore, “age” was also correlated with the trail-making subtests (flexibility, inhibition, working memory and executive attention) “color trail” (ϱ = 0.22, *p* = 0.012) and “gray trail” (ϱ = 25, *p* = 0.029). Hence, ANCOVAs with “age” as a covariate and Kruskal–Wallis tests were conducted to determine between group differences on these variables controlling for age (residual distribution and variance equality were previously checked). ANCOVA revealed significant effects of “group” on “general ability index” after controlling for age, *F* = 4.486, *p* = 0.037, η^2^ = 0.043 (Kruskal–Wallis test, χ^2^ = 4.365, *p* = 0.037). No more differences were found for the rest of the variables.

### 3.2. Relationship between ADHD Symptoms, EF Performance and Cognitive Profile

The possible relationship between ADHD symptoms and EF performance was explored. First, correlational analyses (Pearson’s, Spearman and partial rank correlations) were conducted with the total sample; then, the analyses were re-run by groups. Similarly, the possible relationship between ADHD symptoms and cognitive profile was also explored. For the total sample, no relationships between ADHD symptoms and EF performance were found (*ps* > 0.05). The only exception was a significant positive relationship between “verbal comprehension index” and “ADHD hyperactivity/impulsivity symptoms” (ϱ = 0.29, *p* = 0.003, and ϱ = 0.24, *p* = 0.017, teacher and family form, respectively). No more correlations were significant.

Results by groups were slightly different. “ADHD hyperactivity/impulsivity symptoms” were significantly correlated with “phonemic fluency” (working memory; ϱ = 0.30, *p* = 0.021, and ϱ = 0.28, *p* = 0.034, teacher and family form, respectively) for the foster group, but not for the control group (*ps* > 0.05). No more correlations were found between ADHD symptoms and EF performance. Regarding the cognitive profile, a significant correlation was found between “ADHD hyperactivity/impulsivity symptoms” (family form) and “processing speed index” (ϱ = 0.24, *p* = 0.017) for the foster group, but not for the control group (*ps* > 0.05). No more correlations were significant.

### 3.3. Relationship between ADHD Symptoms, EF Performance and Cognitive Profile with Placement History

After controlling for children’s age, some EF tasks were related to the placement history. Longer stays with the foster family were significantly correlated to higher scores in “Interference” (inhibition) and the trail-making subtests (flexibility, inhibition, working memory and attention), “gray trail”, and “color trail” (Table 2). Moreover, longer stays with the biological family and higher age at entry into foster care were significantly correlated to lower scores in “gray trail” and “color trail” (flexibility, inhibition, working memory and attention).

To conduct the multiple regression analyses, the Z-scores for non-normally distributed variables were calculated. EF and cognitive scores were the dependent variables; ADHD symptoms and placement history were included as the independent variables (age at assessment was also included in the models). The first analysis was a multiple linear regression to predict “verbal comprehension index” based on ADHD symptoms and placement history. The regression equations were not significant (for ADHD symptoms teacher form, *F*(5,53) = 0.517, *p* = 0.763, *R*^2^ = 0.05; for ADHD symptoms family form, *F*(5,53) = 0.439, *p* = 0.819, *R*^2^ = 0.04). No one of the other models conducted was significant. In other words, placement history variables or ADHD symptoms predicted neither EF performance nor cognitive profile (controlling for children’s age).

## 4. Discussion and Conclusions

The first goal of this study was to analyze the ADHD symptomatology, EF performance and cognitive profile of children aged 6–12 years diagnosed with ADHD who were in family foster care than a group of children also diagnosed with ADHD but not involved with protection services. Contrary to our initial hypothesis, ADHD symptoms, executive and cognitive profiles were quite similar in both groups after controlling for children’s age. The only significant difference was that children in foster care had lower GAI scores compared to the control group. Scarce research is available, but several hypotheses can be raised. Previous literature has documented common inattention and hyperactivity symptoms in children in foster care, due in part to many pre-existing factors, including genetic vulnerability, poor socioeconomic conditions, abuse, and violence (see [38]). Some pre-existing factors could exacerbate the inattention and overactivity difficulties of ADHD-diagnosed children in foster care, especially in school contexts, where there are higher attention requirements. The lower GAI—verbal and perceptual skills—scores obtained by the foster group could be due to neglect or lack of stimulation from the birth family or difficulties in schooling caused by multiple placements. Both ADHD groups showed a specific cognitive profile in the WISC-IV. This tool has been widely used to identify cognitive patterns in different neuropsychological disorders [56]. Scores in CPI (working memory and processing speed indices) were classified as low-average than community control scores. In contrast, GAI scores (verbal and perceptual skills) and the full-scale IQ were classified as average. These results match with the cognitive profile found in children diagnosed with ADHD [42,57,58]. Moreover, previous literature has documented a cognitive profile in children in foster care characterized by low results in verbal comprehension and processing speed [31], verbal comprehension and working memory [59], or generalized impairment in all indices [60]. The comparison between previous research and the current study seems to point to different cognitive profiles. Therefore, future lines of research could include a sample of children without ADHD living in foster families.

Furthermore, in relation to EF performance, both groups showed low scores in some EF tasks color trail (flexibility, inhibition, working memory and attention), phonemic fluency (working memory), and Interference (inhibition), but average scores in others semantic fluency (working memory) and gray trail (flexibility, inhibition, working memory and attention) compared to community controls [43]. A similar pattern of difficulties has been found in a local sample of ADHD-diagnosed children [61]. The extant literature on verbal fluency has shown inconclusive results. However, phonemic fluency tended to discriminate somewhat better between ADHD and controls than semantic fluency [62,63]. Previous studies have shown that ADHD-diagnosed children had higher difficulties in inhibition using the Stroop task and shifting through the trail-making test (e.g., [63,64]) compared to community controls, supporting the results reported here.

Our second analysis was focused on the relation between inattentive/hyperactive symptoms, EF performance and cognitive profile. In the foster group, only higher hyperactivity/impulsivity symptoms were related to better phonemic fluency and processing speed. A possible explanation could be that those children in foster care perceived as more hyperactive and impulsive could have a higher processing speed and also perform better in certain tasks, such as phonemic fluency, that push the child to think quickly and to elaborate as many responses as possible. The low number of correlations is consistent with previous research that did not find a relationship between neuropsychological measures of EF and ADHD symptoms [37], nor between parent reports and the children’s intellectual performance [28]. Previous studies have shown that parent reports did not correlate to direct EF tasks but showed strong correlations with parental reports of ADHD symptoms, indicating biases in reporting—same method, same informant [65].

The second goal was to study the relationship between placement variables and ADHD symptoms, EF performance and cognitive profile. Contrary to previous research that has reported weak relations between early placement variables and executive and intellectual capacity in children in foster care [26,28,37] but, following the initial hypothesis, age at placement and time in the foster care family were related to EF outcomes. However, non-significant correlations were found concerning the length of stay in residential care. Children in foster care that were exposed to shorter stays in their biological families and in residential care, and therefore, joined their foster families at an earlier age and had remained longer with them showed better EF performance. This could indicate the beneficial effect on executive and cognitive performance and inattentive/hyperactive behavior of growing up in a stable and nurturing family after experiences of early adversity, as reported in previous studies with adoptive families [35]. These results are not consistent with Wretham and Woolgar [37] with UK children adopted from care, where older age at adoption was related to fewer EF difficulties. However, that result could reflect the fact that children were removed from birth families at an older age due to less severe forms of maltreatment. However, differences in child welfare services across UK and Spain could also partially explain the differences. Nonetheless, longer exposures to less severe adversity can potentially have a detrimental impact on EF development, favored by the protracted development of the prefrontal cortex and its high sensitivity to stressful experiences [66]. Despite the significant correlations between placement history and EF, regression analysis showed that placement variables did not predict EF and intellectual performance after controlling for children’s age. A possible explanation could be that placement variables were distributed in a high range of scores, leading to a lack of statistical power in the regression analysis. Moreover, different variables related to placement trajectories, experiences of abuse and neglect, specific experiences or personal characteristics, which are very difficult to assess in these types of populations, could have a relevant impact on the cognitive and behavioral performance of children in foster care.

## 5. Limitations, Future Lines of Research and Practical Implications

This study had a number of limitations. The sample size was small from a statistical point of view. Still, it is similar to previous studies with this type of specific population that is difficult to access. The local sampling of the foster group may limit the generalization of the findings. Still, the group characteristics were similar to the Spanish population of children in foster care regarding age, but not gender [2], and also similar to the population of ADHD-diagnosed children. In addition, EF was evaluated directly in the children, not through caregivers or teacher reports, which could offer a different perception of the child’s EF in daily activities like school, family and social contexts.

Future lines of research would include a more global approach to other areas of development, such as emotional development and attachment variables, mental health and school achievement. Other developmental stages, such as adolescence, where the executive and cognitive capacities continue to develop academic and social demands, have increased complexity. Another future line of research would be to add a sample of children in foster care with behaviors mimicking symptoms associated with ADHD but without a diagnosis of ADHD. This could help to disentangle the similarities and differences between primary ADHD and ADHD-like symptoms derived from emotional problems or from the experience of traumatic situations [24]. It would be interesting to delve further into the children’s early experiences of neglect and abuse to disentangle their role in the etiology of ADHD in children in foster care. Finally, since some studies have reported different diagnosis rates depending on ethnic/racial characteristics [67], a future line of research could include children from an ethnic/racial background other than Hispanic.

The practice implications of this study involve different groups. On one hand, foster parents and professionals must be aware that these children are more likely to present inattention, hyperactivity and impulsive behaviors, which may be a consequence of an alteration of the neurodevelopment associated with the child’s history. Regarding clinical assessment, ADHD diagnostic guidelines should reflect these particular characteristics of children in foster care, recommending that the higher prevalence of ADHD in this group be considered. Furthermore, following the indications of the DSM-V, a differential diagnosis should be performed to rule out that the symptoms are not better explained by other reasons unrelated to ADHD. Although the difficulties shown by these ADHD-diagnosed children in foster care were not predicted by their placement trajectories, the younger entry and longer exposure to a stable and nurturing family were related to better executive performance, pointing to the beneficial effects of protection policies based on out-of-home family care.

## Figures and Tables

**Table 1 children-08-00405-t001:** Comparisons between children in foster care and control groups for demographic information, ADHD symptomatology, cognitive and executive performance.

Variables *M* (*SD*)		Foster Group	Control Group	χ^2^/*t/U* ^†^	*p*	Effect Size
*n*		59	43			
Gender *n* (% male)		47 (79.7)	34 (79.1)	0.005	0.566	
Age (years)		8.29 (1.92)	8.30 (2.0)	0.04	0.971	
ADHD combined presentation *n* (%)		37 (62.7)	22 (51.2)	1.36	0.311	
ADHD predominantly inattentive presentation *n* (%)		22 (37.3)	21 (48.8)			
ADHD symptomatology (ADHD-RS-IV)						
	Teacher form (raw scores)					
	Inattention	7.64 (1.41)	7.00 (1.23)	1.600	0.018	0.26 ^‡^
	Hyp/imp	4.78 (3.55)	3.51 (3.21)	1.527	0.077	
	Family form (raw scores)					
	Inattention	7.76 (1.36)	7.42 (1.50)	1.427	0.265	
	Hyp/imp	4.92 (3.30)	4.35 (3.09)	1.390	0.411	
Cognitive performance (WISC-IV; IQ scores)						
	Verbal comprehension	104.0 (11.2)	108.8 (13.6)	1.959	0.053	
	Perceptual reasoning	98.9 (12.6)	101.3 (12.2)	0.987	0.330	
	Working memory	86.2 (11.8)	91.2 (14.5)	1.925	0.057	
	Processing speed	91.7 (12.6)	93.9 (12.8)	0.858	0.393	
	General ability index	99.7 (12.2)	105.0 (12.5)	2.126	0.036	0.43 ^§^
	Cognitive proficiency index	86.4 (12.6)	89.0 (13.9)	0.970	0.334	
	Full Scale IQ	94.3 (10.8)	98.4 (12.2)	1.762	0.081	
Executive performance (ENFEN; sten scores)						
	Phonemic fluency	3.78 (1.86)	4.40 (1.94)	1.021	0.090	
	Semantic fluency	5.64 (2.02)	6.44 (2.18)	0.984	0.051	
	Gray trail	4.53 (2.56)	4.91 (2.55)	1.155	0.438	
	Color trail	2.92 (1.97)	3.12 (1.88)	1.170	0.494	
	Interference	3.70 (1.94)	3.70 (1.70)	1.218	0.731	

*Note*. ADHD-RS-IV = ADHD rating scale IV; WISC-IV = Wechsler Intelligence Scale for Children, Fourth Edition; ENFEN = neuropsychological assessment for executive functions in children; Hyp/imp = hyperactivity/impulsivity. ^†^ χ^2^ was used for gender and ADHD percentages; Student’s *t* for WISC variables; and Mann–Whitney *U* for age, ADHD-RS-IV and ENFEN; ^‡^ rank biserial correlation; ^§^ Cohen’s *d.*

**Table 2 children-08-00405-t002:** Correlations between executive function outcomes and placement history.

EF Outcomes	Time with Biological Family	Age at Entry into Residential Care	Age at Placement into Foster Family	Time with Foster Family
Gray Trail	−0.33 *	−0.32 *	−0.36 *	0.36 **
Color Trail	−0.34 **	−0.34 **	−0.44 ***	0.43 ***
Interference	−0.27 *	−0.05	−0.04	0.26 *

* *p* < 0.05; ** *p* < 0.01; *** *p* < 0.001.

## Data Availability

The data that support the findings of this study are openly available in “figshare” at https://doi.org/10.6084/m9.figshare.10288715.v1.

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
