# Peer review of "ADHD Symptomatology, Executive Function and Cognitive Performance Differences between Family Foster Care and Control Group in ADHD-Diagnosed Children"

_children, 2021, doi:10.3390/children8050405_

Round 1
Reviewer 1 Report
Introduction
References should be included to validate certain claims:
Line 35 ff., Line 49 ff.
Line 92 ff.: “…and almost a quarter of the children presented difficulties of clinical significance according to caregivers’ reports”
The amount of “a quarter” should be specified when mentioning a ratio including the description of the sample and methodology used in this research.
Line 111 f.: Should be “foster care”
Participants
Line 159 f. :
“The entire sample of the foster group suffered some type of abuse and/or neglect when they were living with their biological families, although these variables were not included in further analysis.”
Please state where this information comes from! And why it is not included in further analysis, since it is an important part of the earlier stated hypothesis of the aetiology of ADHD in children in foster care. As also stated in the discussion, this should be investigated further.
The authors should specify what kind of comorbidities were assessed, being an important factor when assessing ADHD in research.
Measures
Please state the reason for not including the Rings task in the ENFEN battery.
Sine there is no data available regarding reliability and validity of the ENFEN battery, especially when excluding the Rings task, additional measures such as internal consistency and/or factor scores of the measured scales within this sample should be included.
Please use the corresponding statistical parameters when reporting values, e.g. α=.90 and α=.86 respectively (line 235). The same holds for lines 276 (d=…), 279 (r=…) and so on.
Table 1
It is not clear, what variables were used to calculate the correlation reported for the scale Inattention (r = .26) – effect sizes for differences between groups should be reported as Cohen’s d or the like, especially regarding the title of the table (Comparisons between…)
Values without and within brackets should be explained, as in the first part of table 1 (% male, years,…)
It is not clear what statistical parameter is used in the fourth column (χ2/t /U†). Furthermore, these values should not be negative, please report the absolute value. The same accounts for effect sizes.
Please delete “This section may be divided by subheadings. It 350 should provide a concise and precise description of the experimental results, their inter-351 pretation, as well as the experimental conclusions that can be drawn.” (lines 350 ff.)
Author Response
We include a pdf document with the answer, point by point, to the questions you ask us. Best regards.
Reviewer 2 Report
In this article entitled “ADHD Symptomatology, Executive Function and Cognitive Performance Differences Between Family Foster Care and Control Group in ADHD-diagnosed Children” the authors investigated the intellectual and executive profile and symptomatology in foster/control group children with ADHD and the role of placement history, EF performance, cognitive profile in the group of children in foster care. The topic has good potential to be of interest for Children readers and the paper is well written. Some minor revisions are needed, to improve its overall quality: Introduction and discussion paragraphs are well written and detailed. Please more clearly underline what gap in the literature this article aims to fill and what are the novelty and peculiarities of the paper. It is not clear if subjects with other diagnoses/comorbidities are included or excluded in the sample. It could be an important bias to consider in discussion or limits. It is not clear the “iii- inclusion criteria”, value six is a raw score or not? Is it a score about the total scale? What represents this value?
Author Response

(The authors gave the same response as above.)
